# MicroRNAs in Medicinal Plants

**DOI:** 10.3390/ijms231810477

**Published:** 2022-09-09

**Authors:** Mingyang Sun, Shiqiang Xu, Yu Mei, Jingyu Li, Yan Gu, Wenting Zhang, Jihua Wang

**Affiliations:** 1Crops Research Institute, Guangdong Academy of Agricultural Sciences, Guangzhou 510640, China; 2Guangdong Provincial Key Laboratory of Crop Genetic Improvement, Guangdong Academy of Agricultural Sciences, Guangzhou 510640, China; 3Guangdong Provincial Engineering and Technology Research Center for Conservation and Utilization of the Genuine Southern Medicinal Resources, Guangzhou 510640, China

**Keywords:** medicinal plant, miRNA, synthesis pathway, biological function

## Abstract

Medicinal plant microRNAs (miRNAs) are an endogenous class of small RNA central to the posttranscriptional regulation of gene expression. Biosynthetic research has shown that the mature miRNAs in medicinal plants can be produced from either the standard messenger RNA splicing mechanism or the pre-ribosomal RNA splicing process. The medicinal plant miRNA function is separated into two levels: (1) the cross-kingdom level, which is the regulation of disease-related genes in animal cells by oral intake, and (2) the intra-kingdom level, which is the participation of metabolism, development, and stress adaptation in homologous or heterologous plants. Increasing research continues to enrich the biosynthesis and function of medicinal plant miRNAs. In this review, peer-reviewed papers on medicinal plant miRNAs published on the Web of Science were discussed, covering a total of 78 species. The feasibility of the emerging role of medicinal plant miRNAs in regulating animal gene function was critically evaluated. Staged progress in intra-kingdom miRNA research has only been found in a few medicinal plants, which may be mainly inhibited by their long growth cycle, high demand for growth environment, immature genetic transformation, and difficult RNA extraction. The present review clarifies the research significance, opportunities, and challenges of medicinal plant miRNAs in drug development and agricultural production. The discussion of the latest results furthers the understanding of medicinal plant miRNAs and helps the rational design of the corresponding miRNA/target genes functional modules.

## 1. Introduction

Since time immemorial, medicinal plants have fully demonstrated their therapeutic potential [1]. Medicinal plants from *Asteraceae* and *Lamiaceae* have significant effects in the treatment of cardiovascular-related diseases, including antioxidative and vasodilative activities [2]. In 2003, many descriptions of medicinal plants were used to prevent and treat severe acute respiratory syndrome (SARS) [3,4]. The clinical trials showed that prescriptions containing *Ephedra sinica*, *Glycyrrhiza uralensis*, and *Rehmannia glutinosa* reduce the severe symptoms of new coronavirus COVID-19 patients [5]. Although the contribution of medicinal plants to the treatment of human diseases is obvious, their cultivation area is not as wide as that of staple crops. Lacking in resources, degradation of the environment, and irregular cultivation patterns still challenge the production of medicinal plants [6]. Therefore, studies of medicinal plants have also centered on breeding high-yielding varieties with adaptability to environmental stress conditions.

MiRNAs, a type of small non-coding RNAs (usually 20–24 nts), have been defined as crucial post-transcriptional gene regulators [7]. Plant miRNAs play important roles in growth and development, primary and secondary metabolism, response to environmental challenges, and exogenous regulation [8]. Following the publication of the first report in 2009, research on miRNA in medicinal plants has been increasing (Figure 1) [9]. The function of medicinal plant miRNAs is divided into cross- and intra-kingdom groups. The cross-kingdom studies of medicinal plant miRNAs concentrate on the major human disease treatments [10]. Six *Ocimum basilicum* miRNAs involving miR160, 414 and 869.1 were found to modulate 26 human malignancy-related target genes [11]. Aba-miRNA-9497 in *Atropa belladonna* was highly homologous to *Homo sapiens* miRNA-378, which both targeted the 3′-untranslated region (3′-UTR) of the mRNA encoding the neurologically relevant, zinc-finger transcription factor ZNF-691 [12]. The *TRAF2* gene, a potential target of miRNAs in *Bacopa monnieri*, is the upstream signaling factor in the cancer pathway [13]. In the intra-kingdom, miRNA research on medicinal plants is mainly focused on the synthetic and metabolic pathways of medicinal secondary metabolites. In an endangered medicinal plant *Picrorhiza kurroa*, miR-4995, miR-5532, and miR-5368 participated in terpenoid biosynthesis and culture growth conditions [14]. Target genes of hop (*Humulus lupulus*) miRNAs responding to viral infection belonged to prenylflavonoid biosynthesis, and growth and development [15]. The comparative genomics approach identified 45 *Aquilegia coerulea* miRNAs that target genes involved in metabolism and stress responses [9]. As of yet, the mature research system on miRNAs in medicinal plants has not been established.

In light of the irreplaceable therapeutic value of medicinal plants and the diverse roles of miRNAs, the biosynthetic pathways and functions of medicinal plant miRNAs are comprehensively classified in this article. The criteria for document search were as follows: (1) database, Web of Science Core Collection; (2) edition, SCI-EXPANDED-1980-present; (3) searched field, All Fields; (4) searched word pairs, medicinal plant + microRNA, herb + microRNA, traditional Chinese medicine + microRNA, and herbaceous plant + microRNA (the “+” means another search line); (5) document types, article; (6) language, English; and (7) searched date, 5 September 2022. A manual search was also used to retrieve relevant papers from the results of the automatic search. The papers not included in Journal Citation Reports (JCR) were deleted, and the remaining papers were selected in this study. MiRNAs in a few medicinal plants with remarkable pharmacological effects have made breakthroughs. Research on other medicinal plant miRNAs is just beginning, but their potential applications have already emerged. Overall, this review summarizes the characteristics of the discipline development of miRNAs in medicinal plants, facilitates the establishment of a complete cross- or intra-kingdom miRNA/target gene module verification system, and discusses the possible challenges of systematic research in the future.

## 2. Biosynthetic Pathways of Medicinal Plant miRNAs

Eukaryotic miRNAs are synthesized based on both canonical and non-canonical mechanisms [16,17]. MiRNAs in medicinal plants were primarily generated from the canonical miRNA-generating pathway, which refers to the dicer-dependent pathway. For the non-canonical pathway, different miRNA origins were recognized in introns [18], rRNAs [18,19], snoRNAs [20], endogenous siRNAs [21], and tRNAs [22]. The atypical miRNAs generated through the precursor rRNA (pre-rRNA) splicing process have been found in medicinal plants [18,23].

### 2.1. Characteristics of the Canonical Pathway

The core promoter region of the miRNA gene has TATA-box and transcription start site (TSS). The TSSs of miRNA genes are mainly located in intergenic regions, introns and reverse complementary sequences of coding sequences [17]. In the nucleus, after transcribed, primary-miRNA (pri-miRNA) having 3′ tail and 5′ cap is formed. One or several internal stem loops are folded from single pri-miRNAs [16]. The length of plant pri-miRNA hairpins is heterogeneous, ranging from approximately hundreds to even thousands of bases [22]. Pri-miRNA hairpins become precursor miRNAs (pre-miRNAs or MIRNAs) when modified by ribonuclease enzyme Dicer-like 1 enzyme (DCL1) [19]. This process is called the first cleavage (Figure 2). The stem-loop of plant MIRNAs has a wider range in length (70 to 350 nts) than that of animal MIRNAs (65 to 70 nts) [24]. In the second cleavage, MIRNA is cleaved into miRNA: miRNA * duplexes (21 to 24 bp) [25]. The two-step cleavage takes place in subnuclear regions called Dicing-bodies [26,27,28].

According to the length of the stem, plant MIRNA hairpins produce one or several duplexes [29]. The duplexes are then methylated with Hua enhancer1 (HEN1) at 3′ ends [30]. The methylation enhances plant miRNA stability by preventing non-template 3′-polymerization that accelerates miRNA turnover [31]. An exportin5 (Exp5) homolog transports the duplex from the nucleus into the cytoplasm. The miRNA* strand (passenger strand) is then degraded, while the mature single-strand miRNA (guide strand) is reserved. MiRNA is carried with Argonautes (AGOs, mainly AGO1), the core protein for RNA-induced silencing (RISC) complex [32,33]. MiRNA guides the RISC complex to mRNA strand via almost complete base complementation, and the RISC endonucleolytic cleavage the mRNA [7]. After binding to target mRNA, a few plant miRNAs do not perform cleavage function but reduce translation efficiency [34,35,36].

Plant miRNA target sites can be found anywhere in transcripts, for instance, 5′-UTRs, open reading frames (ORFs), 3′-UTRs, and noncoding transcripts. This finding suggests that all RNA environments are equally suitable for miRNA regulation in plants. Although animal miRNAs conservatively bind in the 3′-UTRs, the number of targets of a given plant miRNA is generally less than that of a given animal miRNA by at least an order of magnitude. This phenomenon is due to the low base complementary requirement of animal miRNAs to the target sites [37].

### 2.2. Discovery of the rRNA-Derived Non-Canonical Pathway

The transcript of pre-rRNA gene *RN45s* in eukaryotes contains 18S, 5.8S, and 28S rRNA regions with two internal transcribed spacers (ITS1 and 2). The sequence of rRNA regions is conserved and is retained completely during the subsequent rRNA splicing process. The ITSs contain more species-specific nucleotide sequences that are used as phylogenetic markers [38]. The first pre-rRNA-derived miRNA (miR-712) was found in the murine ITS2 region, in which the mature body was modified without the conventional dicer-dependent manners. The miR-712 alleviated atherosclerosis via regulating the tissue inhibitor of metalloproteinase 3 (TIMP3) [39].

There are also rRNA-derived miRNAs found in *Papaver somniferum* and *Lonicera japonica*. A clustered *P. somniferum* miRNA site was identified in a long polycistronic pre-rRNA region. Five of these clustered miRNAs were species conserved, which MIRNA sequences were highly homologous to 13 plant species [18]. This result suggests that the clustered *P. somniferum* miRNA site may not be located in ITS regions. Interestingly, such clustered miRNAs showed similar conserved pattern is rare in plants [40,41], but common in animals [42,43,44,45]. The silencing mechanism of pre-rRNA-generating miRNA remains unclear in medicinal plants, reports have shown their functional importance. Oral administration of honeysuckle decoction (HD) prevented influenza A viruses (IAVs) infection and decreased H5N1-induced mouse death due to the rRNA-derived MIR2911 (precursor of miR2911) in HD [23]. The molecular mechanism of medicinal plant-derived non-canonical miRNAs still has extensive worthwhile space for exploration.

## 3. Functional Research Progress of Medicinal Plant miRNAs

### 3.1. Cross-Kingdom Regulation

MiRNAs biosynthesized from medicinal plants can act as botanicals to regulate health-related processes. Oral plant-based diets will enable the cross-kingdom transfer of miRNAs in medicinal plants into mammalian circulation. In Table 1, miRNAs of eight medicinal plants have been predicted or experimentally proven to be bioactive factors for human disease treatment.

#### 3.1.1. Exceptional Stability

Unlike RNAs degraded during drug processing, medicinal plant miRNAs functional in cross-kingdom regulation are robust during soaking, boiling and homogenization processes. These miRNAs survive even under adversely stable animal systemic circulation, such as extreme pH (simulated gastric juice at pH 1.2), bowel movements, and ribonuclease (RNase) treatment [147]. A significant increase in an atypical rRNA-derived MIR2911 in both blood and urine was first discovered in mice that were fed with a diet of honeysuckle for several days [23]. Thousands of miRNAs derived from 10 medicinal plants were transferred to human blood cells and tissues following oral herbal decoctions (Table 1) [54]. 

Research showed that the stability of exogenous miRNAs during drug preparations was associated with the protection of plant macromolecules [55]. Through mammalian dietary uptake, miRNAs self-assembled into exosomes and were transported into the circulation and target tissues or cells [148,149,150]. Furthermore, there are several other factors to enhance the stability of medicinal plant miRNAs in mammals: (1) the 2′-O-methylations protect plant miRNAs avoiding degradation of exonucleolytic digestion and uridylation [7,31]; (2) high G, C content, sturdy structure and absence of RNases digestion motifs of miRNAs guarantee the stability [151]; (3) the miRNA-binding proteins like argonaute proteins (AGOs) [152] and nucleophosmin 1 [153] prevent circulating miRNAs from decay; and (4) medicinal plant metabolites create an environment for miRNAs to inhibit RNase activity [154].

The above conditions ensure the reliability of medicinal plant miRNAs in clinic treatment. However, doubts remain. A few researchers claim that plant-derived miRNAs are almost undetectable in plant-fed animal bodies [155,156]. There are five reasons for this phenomenon: (1) the diversity of sequences, structures and binding proteins are the fundamental reasons for the differences in the stability and biological activity of medicinal plant miRNAs [157]; (2) medicinal plant miRNA can effectively inhibit the function of target mammalian mRNA only when it has more than 100 copies per cell [158]; (3) the protective methylated structure adds difficulty to the identification of medicinal plant miRNAs; (4) medicinal plant miRNAs may not be detected in plasma or tissues when an herbal diet is fed for a short time and in small doses; and (5) health condition of the digestive system directly affects the absorption efficiency of medicinal plant miRNAs [159]. Accordingly, standardized plant-specific exogenous miRNA detection technology and experimental design are the prerequisites to increase the accuracy of cross-kingdom medicinal plant miRNA research [160].

#### 3.1.2. Targeting Genes Associated with Major Diseases

The cross-kingdom function of medicinal plant miRNAs has been experimentally verified to be unexpectedly strong (Table 2). In addition to inhibiting H5N1 and H7N9 viral activity in vitro and in vivo, MIR2911 also suppressed the replication of H1N1 by decreasing H1N1-encoded PB2 and NS1 protein expression [23]. In 2019, the sudden outbreak of COVID-19, of which the causative virus is the syndrome coronavirus 2 (SARS-CoV-2), challenges the safety of human life to date [161]. MIR2911 absorbed by COVID-19 patients can promptly and effectively inhibit SARS-CoV-2 replication via binding 179 candidate target sites in the SARS-CoV-2 transcriptome [46]. In cancer treatment, the miR2911 strongly bound and down-regulated the expression of TGF-β1, retarding the colon cancer process with an increase in T lymphocyte infiltration in mice [49].

The NF-kB protein family in animals is a well-known anti-inflammatory and immune regulator. *Gastrodia elata* (GE) is a precious herbal medicine, in which miRNAs have been identified through the Illumina platform. Cell transfection showed that Gas-miR01 and Gas-miR02 of GE prominently restrained the accumulation of *Homo sapiens* A20 protein driven by NF-kB [48]. In addition to MIR2911 and Gas-miR01/02, the cross-kingdom functional exploration of medicinal plant miRNAs remains at the stage of computer prediction.

Site accessibility, low free energy, and base-pairing between the “seed” region of miRNA and target gene are features generally used for the computational recognition of heterogenous targets [162]. Xie (2017) predicted human target genes of miRNAs in *Viscum album* combined with four frequently used animal target prediction algorithms (TargetScan, miRanda, PITA, and RNAhybrid) [51]. Korean ginseng (*Panax ginseng*) has been commonly and efficiently used as medicine for thousands of years. Numerous disease-related target genes of Korean ginseng miRNAs were found combining RNAhybrid, miRanda, and TargetScan [50]. The medicinal plant happy tree (*Camptotheca acuminata*) is a deciduous tree having anticancer properties. A total of 152 human target genes associated with prominent types of cancers were predicted to be regulated by 14 highly stable putative novel miRNAs in the happy tree (Table 1) [47]. Although, the therapeutic role of a few medicinal plant miRNAs in mammalian major diseases has been well demonstrated [23,163], the potential mining of most medicinal plant miRNAs involved in human health regulation is still at an early stage.

### 3.2. Intra-Kingdom Regulation

In addition to miRNAs, the unique secondary metabolites of medicinal plants also exhibit abundant pharmacological activities. In the medicinal plant kingdom, a large number of miRNAs have been found to participate in the biosynthesis of secondary metabolites [61]. Consistently, the literature statistics in the present study showed that research on medicinal miRNAs revolved around the secondary metabolite synthesis pathway, followed by the growth, development, and environmental stress response pathways (Table 1 and Table 2).

#### 3.2.1. Secondary Metabolism

The synthesis process of secondary metabolites in medicinal plants is complicated [164]. The transgenic plant lines are important experimental materials for the selected miRNAs and target genes functional studies. In patchouli (*Pogostemon cablin*), miR156-targeted squamosa promoter binding protein-like (SPL) transcription factor plays a crucial role in the patchouli oil (largely composed of sesquiterpenes) accumulation. Yu et al. (2015) demonstrated the regulatory effect of the miR156-SPL-PTS (patchoulol synthase) module on patchouli oil production by testing SPL10 and MIR156 transgenic patchouli lines [56]. The miR408 also acts in the regulation of secondary metabolism. In the medicinal model plant *Salvia miltiorrhiza*, recombinant laccase LAC3 was verified to be the target of Sm-miR408 using 5′-rapid amplification of cDNA ends (5′-RACE). The contents of salvianolic acid B (SalB) and rosmarinic acid (RA) were induced in both miR408-lacked and SmLAC3-overexpressed transgenic *S. miltiorrhiza* lines (Table 2) [62].

Cultivation of transgenic hairy roots is a common method for miRNA functional research on secondary metabolites in *S. miltiorrhiza*. miR396 is conserved and plays various roles in plants. MiR396b-overexpressing *S. miltiorrhiza* hairy roots repressed the hairy root growth and salvianolic acid concentration, but induced the tanshinone accumulation [68]. Further verification indicated that SmGRFs, SmHDT1, and SmMYB37/4 were targets of *S. miltiorrhiza* miR396b, which mediated the gibberellin signaling pathways and consequentially resulted in phenotype variation [68]. Conversely, the ath-miR160a overexpressed *S. miltiorrhiza* hairy roots down-regulated the expression levels of ARF10, 16, and 17, inhibited the biosynthesis of tanshinones, and increased hairy root biomass (Table 1 and Table 2) [61]. These results reflect the complexity of the effects of miRNAs on the mechanism of tanshinone synthesis.

The overexpression of miR8154 and miR5298b in the Taxus cell line upregulated the major enzyme genes related to taxol, phenylpropane, and flavonoid synthesis [64]. In Table 1, there are many other studies that combined high-throughput sequencing and bioinformatics to identify medicinal plant miRNAs and predict target pathways. 

#### 3.2.2. Growth and Development

Studies have exhibited that the spatiotemporal specificity expression of medicinal plant miRNAs plays a pivotal role in growth and development [165]. Zeng et al. (2015) identified the miRNA levels in *Lycium barbarum* seedlings at four developmental stages (S1-S4) using Illumina HiSeq^TM^ 2000 platform [106]. Functional prediction of differentially expressed miRNAs revealed the characteristics of fruit ripening miRNA-mediated mechanism. The *Hypericum perforatum* flowers shared highly conserved miRNAs and these miRNAs potentially target functional genes involved in stress response, flower development, and plant reproduction [114]. The production and degradation of secondary metabolites are usually organ- and tissue-specific, and their accumulations and compositions change during plant germination, development and aging [166]. The tissue-specific expressing characteristic of a total of 232 miRNAs containing four tissues in *Opium poppy* was comprehensively performed using miRNA microarray technology [57]. Target gene functional prediction of these miRNAs revealed that some miRNAs might be involved in quinoline alkaloids (BIA) biosynthesis, including pso-miR13, pso-miR2161 and pso-miR408 (Table 2). In *Ginkgo biloba* (the “living fossil” in plant), 3314 miRNAs were identified from five organs using northern blot, quantitative real-time PCR (qRT-PCR), RACE, and degradome sequencing. Among them, four conserved miRNAs and five novel miRNAs might participate in terpene trilactones (TTL) biosynthesis pathways by targeting 12 predicted TTL biosynthesis genes (Table 1) [65].

A study of honeysuckle selected the suitable reference miRNA genes for the quantification of target miRNA expression through tissue- and variety-specific qRT-PCR [107]. Data of cycling threshold (Ct) value ranges of qRT-PCR and algorithms from GeNorm, NormFinder, and RefFinder were employed. Two stable honeysuckle miRNA reference genes (u534122 and u3868172) were found in three tissues of 21 cultivars from 16 origins. 

#### 3.2.3. Environmental Stress Response

Under biotic or abiotic stresses, medicinal plants develop numerous miRNA-regulate mechanisms to adapt to environmental challenges (Table 1). The plant sequence-conserved miR408 participates in various stress responses. The β-glucuronidase staining results of transgenic tobacco lines expressing *Sm-MIR408pro::GUS* revealed that *Sm-MIR408* is a positive response factor to salt stress. Further study showed that tobacco lines expressing *Sm-MIR408* increased the seed germination rate and decreased the accumulation of reactive oxygen species [6]. The biotic stress ascochyta blight (AB) limits chickpea (*Cicer arietinum*) production worldwide. Chickpea seedlings involved two susceptible genotypes, two resistant genotypes, and an introgression line containing two AB-resistance quantitative trait loci were treated infected with or without AB at two time points. The differentially expressed gene analysis combined with miRNA and mRNA transcriptome from these genotypes totally evaluated 12 miRNA-mRNA regulatory modules [127]. In an economic medicinal tree *Aquilaria sinensis*, expression pattern analysis indicated that eight miRNAs were wound-responsive during the recovery process after wound treatment. One miRNA was identified to be the miRNA * of asi-miR408 in *A. sinensis*, but with the accumulation greatly exceeding that of asi-miR408, suggested that it may have a biological function [133].

#### 3.2.4. Other Fields

Additionally, research on medicinal plant miRNAs also refers to the evolutionary characteristics and database development, etc. (Table 1). Cardamom (*Elettaria cardamomum*) has been used as ayurvedic medicine for a long history. There are 1168 and 1025 unique potential targets of miRNAs found in wild and cultivated cardamom, respectively [118]. Many compounds of the genus *Hypericum* have therapeutic potential, and the miRNA expression profile of five *Hypericum* species was characterized in silico. The miR168 was identified only in *H. perforatum* and *H. kalmianum* with one highly conserved target gene (protein AGO1-like) [145]. In 2019, the first medicinal plant miRNA database, MepmiRDB (http://mepmirdb.cn/mepmirdb/index.html, accessed on 16 April 2018), was published, consisting of miRNA information on sequences, expression levels and regulatory networks from 29 species [144]. 

## 4. Conclusions

Natural drugs of medicinal plants are less toxic, have fewer side effects, and are cheaper and more available to patients around the world [167]. These pharmacologically active components are characterized by multiple molecular types, although most studies have focused on secondary metabolites at the metabolic level. As more reports unraveled the role and significance of medicinal plant miRNAs in tumor proliferation and virus replication, it is evident that they must be novel regulators of human disease at the post-transcriptional level (Figure 3). Initial research indicated that medicinal plant-derived miRNAs in a stable form can resist not only the preparation process, but the digestion and circulation process, ensuring their disease regulatory activity [47]. Nanoparticles mainly contain plant miRNAs, lipids, and proteins that have been demonstrated as natural therapeutical drugs for cancers and inflammatory diseases [168]. However, some reports related to the cross-kingdom function of miRNAs in medicinal plants remain only software predictions, the conclusions of which lack credibility. The differences between individuals also complicate the verification of cross-kingdom functionality. To prevent the question of authenticity in the academic community, here are four suggestions for future cross-kingdom research of medicinal plant miRNA. First, the stability of medicinal plant miRNAs needs to be systematically validated in vivo and in vitro. Second, reproducible validation of plant-miRNA/animal-target modules in human disease treatment should be carried out on candidate miRNAs that exist in at least 100 copies per cell. Third, multidimensional observation should be arranged for the effects of miRNAs on mice and patients with different health levels. Finally, it is also important to assess potential risks before clinical treatment, as a single miRNA can target multiple target genes.

Before the discovery of cross-kingdom functions, medicinal plant miRNAs have shown important regulatory capabilities in the synthesis and modification of metabolites, seedling development, and stress adaptation in the plant kingdom (Figure 3). Methods of intra-kingdom identification and expression analysis of medicinal plant miRNAs mainly include direct cloning, expressed sequence tags analysis and transcriptome sequencing. Bioinformatics, degradome sequencing, qRT-PCR, 5’-RACE, northern blot, transient luciferase, and GUS technologies are used to determine target genes. Unfortunately, after the prediction or indirect verification of the miRNA-target gene modules in medicinal plants, many studies have not conducted in-depth functional research. Only about 10% of studies have reached phenotypic conclusions by developing transgenic seedlings, tissues, or cells. There are probably three reasons: (1) researchers pay far less attention to medicinal plants than to food crops, resulting in many medicinal plants not yet established as efficient genetic transformation systems; (2) the quality of medicinal plants is strongly dependent on the edaphic and climatic environments, which severely restrict their large-scale cultivation; and (3) natural secondary metabolites like terpenes, flavonoids, alkaloids, phenols, etc. which are easier to extract and preserve than miRNAs, leading the researchers to focus on their pharmacological effects first. Nonetheless, the intra-kingdom functional validation of medicinal plant miRNAs has entered a new era. The technology of genome assembly in non-model plants is becoming mature. In the last three years, more than 100 articles (over 60%) have been published on genomic information about medicinal plants [169]. Some medicinal plants have published two or more genome versions (e.g., *Panax notoginseng* [170,171,172,173,174], *Andrographis paniculata* [175,176], and *Gastrodia elata* [177,178]), which have significantly supported the miRNA research. The research progress of medicinal plant miRNAs will develop rapidly in the next few years. With the enrichment of the cross-kingdom function of miRNAs derived from medicinal plants, their intra-kingdom influence will also be given enough attention. More and more in-depth experiments will be used to provide molecular biology evidence of the predicted candidate miRNA/target gene modules.

Since miRNAs were discovered in medicinal plants, studies of their biogenesis and mechanism of action have achieved several milestones (Figure 1). Their function extended from the plant kingdom to the animal kingdom (Figure 3). Despite the promising data thus far, more efforts are necessary to understand the mechanisms. As summarized above, the application potential of medicinal plant miRNAs is enormous, both in terms of improving the yield and quality of source species and in terms of molecular regulation in the treatment of human diseases. 

## Figures and Tables

**Figure 1 ijms-23-10477-f001:**
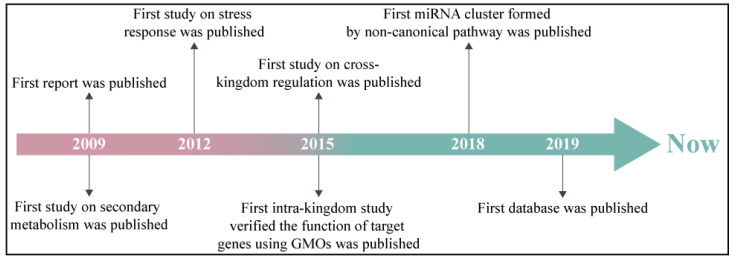
Historical process of miRNA research in medicinal plants. GMOs—genetically modified organisms.

**Figure 2 ijms-23-10477-f002:**
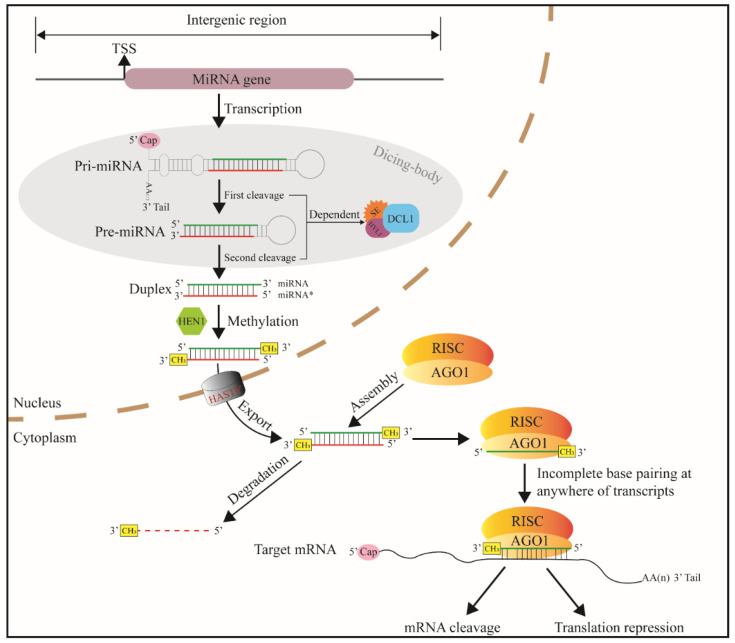
Canonical miRNA-generating pathway in plant kingdom. TSS—transcription start site, RISC—RNA-induced silencing complex, AGO1—argonaute protein, DCL1—Dicer-like 1, SE—C2H2-zinc finger protein serrate, HYL1—Hyponastic leaves 1, and miRNA*—passenger strand of the miRNA: miRNA* duplex.

**Figure 3 ijms-23-10477-f003:**
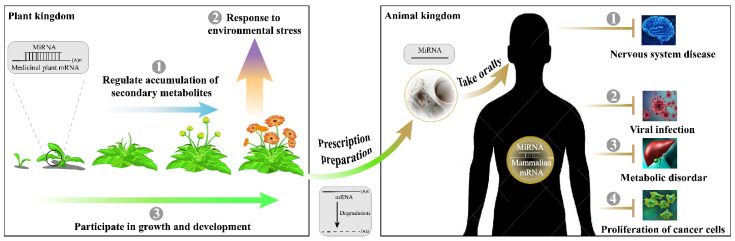
Multifunctional role of miRNAs in medicinal plants.

**Table 1 ijms-23-10477-t001:** Summary of medicinal plant miRNA studies of different species.

Latin Name	Aim Pathway	Methods of Target Functional Identification			References
		Prediction	Indirect Verification	GMOs/GMCs Direct Validation	
Cross-kingdom regulation					
*Lonicera japonica*	Replication of influenza A virus	−	−	√	[23]
*Lonicera japonica*	Replication of COVID-19	−	−	√	[46]
*Camptotheca acuminata*	Breast cancer, leukemia and lung cancer	√	−	−	[47]
*Gastrodia elata*	Homo sapiens A20 gene	−	−	√	[48]
*Ocimum basilicum*	Rheumatoid arthritis and diabetes mellitus	√	−	−	[11]
*Lonicera japonica*	Tumor proliferation	−	−	√	[49]
*Atropa belladonna*	Central nervous system toxicity	−	√	−	[12]
*Panax ginseng*	Cancers, immune diseases, and neurological disorders	√	−	−	[50]
*Viscum album*	Cancers	√	−	−	[51]
*Ocimum basilicum*	Cardiomyopathy, HIV, Alzheimer’s diseases and cancers	√	−	−	[52]
*Bacopa monnieri*	NF-kB and MAPK pathways	√	−	−	[13]
*Aucklandia lapp*, *Rhodiola crenulata*, and *Taraxacum mongolicum*	Stability assessment of miRNAs during decoction preparation	−	−	−	[53]
Ten medicinal plants	MiRNAs were detected in mammalian blood and tissues	√	−	−	[54]
*Viscum album*	Stability assessment of miRNAs during decoction preparation	−	−	−	[55]
Intra-kingdom secondary metabolism					
*Pogostemon cablin*	Synthesis of sesquiterpenes	−	−	√	[56]
*Papaver somniferum*	Benzylisoquinoline alkaloid synthesis	−	√	−	[57]
*Artemisia annua*	Artemisinin synthesis	√	−	−	[58]
*Euphorbia kansui*	Terpenoid biosynthesis	√	−	−	[59]
*Glycyrrhiza*	Glycyrrhizic acid synthesis	√	−	−	[60]
*Salvia miltiorrhiza*	Tanshinone synthesis and biomass	−	−	√	[61]
*Salvia miltiorrhiza*	Synthesis of salvianolic acid	−	−	√	[62]
*Podophyllum hexandrum*	Podophylloxin synthesis	−	√	−	[63]
*Taxus*	Taxol, phenylpropanoid, and flavonoid biosynthesis	−	−	√	[64]
*Ginkgo biloba*	Terpene trilactone synthesis	−	√	−	[65]
*Desmodium styracifolium*	Schaftoside biosynthesis	√	−	−	[66]
*Camellia sinensis*	Catechin, theanine and caffeine synthesis	√	−	−	[67]
*Salvia miltiorrhiza*	Tanshinone, salvianolic acid, and biomass	−	−	√	[68]
*Catharanthus roseus*	Terpenoid indole alkaloids	−	√	−	[69]
*Hippophae rhamnoides*	Lipid synthesis	−	√	−	[70]
*Artemisia annua*	Artemisinin synthesis	−	√	−	[71]
*Salvia miltiorrhiza*	Phenolic acid synthesis	−	√	−	[72]
*Camellia sinensis*	Catechin synthesis	−	√	−	[73]
*Picrorhiza kurroa*	Terpenoid synthesis	−	√	−	[14]
*Dendrobium nobile*	Synthesis of dendrobine	√	−	−	[74]
*Digitalis purpurea*	Cardiac glycoside biosynthesis	−	√	−	[75]
*Panax notoginseng*	Synthesis of triterpenoid saponins	−	√	−	[76]
*Lycoris aurea*	Alkaloid synthesis	−	√	−	[77]
*Acacia*	Lignin and flavonoid synthesis	√	−	−	[78]
*Murraya koenigii*	Flavonoid and terpenoid synthesis	√	−	−	[79]
*Catharanthus roseus*	Secondary metabolism	√	−	−	[80]
*Salvia sclarea*	Phenylpropanoids and terpenoids synthesis	√	−	−	[81]
*Zingiber officinalis*	Gingerol synthesis	√	−	−	[82]
*Ocimum basilicum*	Secondary metabolism	√	−	−	[83]
*Taxus chinensis*	Taxoid synthesis	√	−	−	[40]
*Ferula gummosa*	Synthesis of ferulide	−	√	−	[84]
*Lycium chinense*	Lycopene synthesis	−	√	−	[85]
*Salvia miltiorrhiza*	Biosynthesis of tanshinones	−	√	−	[86]
*Xanthium strumarium*	Terpenoid biosynthesis	√	−	−	[87]
*Salvia miltiorrhiza*	Phenolic synthesis	−	√	−	[88]
*Azadirachta indica*	Secondary metabolism	√	−	−	[89]
*Withania somnifera*	Withanolide synthesis	−	√	−	[90]
*Mentha*	Essential oil biosynthesis	√	−	−	[91]
*Salvia miltiorrhiza*	Tanshinone Synthesis	−	−	√	[61]
*Artemisia annua*	Artemisinin synthesis	√	−	−	[92]
*Vinca minor*	Synthesis of terpenoid indole alkaloids	√	−	−	[93]
*Curcuma longa*	Curcumin biosynthesis	√	−	−	[94]
*Podophyllum hexandrum*	Podophyllotoxin synthesis	−	√	−	[95]
*Podophyllum hexandrum*	Podophyllotoxin synthesis	−	√	−	[96]
*Persicaria minor*	Terpenoid and GLV synthesis	−	√	−	[97]
*Gleditsia sinensis*	Synthesis of monoterpenes and alkaloids	√	−	−	[98]
*Glycyrrhiza uralensis*	Secondary metabolism	√	−	−	[99]
*Capsicum annuum*	Anthocyanin synthesis	√	−	−	[100]
*Brassica oleracea*	Secondary metabolism	−	√	−	[101]
*Persicaria minor*	Terpenoid and GLV synthesis	−	√	−	[102]
*Dryopteris fragrans*	Terpenoid synthesis	−	√	−	[103]
*Echinacea purpurea*	Anthocyanin biosynthesis	√	−	−	[104]
Intra-kingdom growth and development					
*Papaver somniferum*	Root, stem, leaf and young capsule prior to flowering tissues	−	√	−	[57]
*Panax ginseng*	Flower buds, leaves, and lateral roots	√	−	−	[105]
*Lycium barbarum*	Different fruit stages	−	√	−	[106]
*Lonicera japonica*	Flower buds, leaves, and stems of 21 cultivated varieties	−	−	−	[107]
*Lycopersicon esculentum* and *Lycium chinense*	Shoot and fruit of grafted tomato	√	−	−	[108]
*Ginkgo biloba*	Roots, stems, leaves, microstrobilus, and ovulate strobilus	−	√	−	[65]
*Camellia sinensis*	Buds, different development stages of leaves and stems	√	−	−	[67]
*Dendrobium officinale*	Flower, root, leaf and stem	√	−	−	[109]
*Panax notoginseng*	Root with various biomasses	−	√	−	[110]
*Carthamus tinctorius*	Seed, leaf, and petal	−	−	−	[111]
*Panax ginseng*	Roots, stems, leaves and flowers	√	−	−	[112]
*Panax notoginseng*	Roots, stems, and leaves of 1-, 2-, and 3-year-old seedlings	−	√	−	[76]
*Gynostemma pentaphyllum*	three stages of developmental stem-to-rhizome transition	−	√	−	[113]
*Hypericum perforatum*	Flower parts	√	−	−	[114]
*Pinellia ternate*	Leaves, stalks and tubers	−	−	−	[115]
*Lonicera japonica*	Flowers including 2 varieties of honeysuckle at 2 locations	−	√	−	[116]
*Ginkgo biloba*	Epiphyllous ovule leaves and normal leaves	−	√	−	[117]
*Elettaria cardamomum*	Cultivar and wild cardamom genotypes	−	√	−	[118]
*Ginkgo biloba*	Mature ovules (pollination stage) and leaves of female trees	√	−	−	[119]
*Ginkgo biloba*	Cambial structure	−	−	−	[120]
*Passifora edulis*	Inter-tissue and inter-varietal	√	−	−	[121]
*Ginkgo biloba*	Female and male leaves	√	−	−	[122]
*Dendrobium officinale*	Conventional and micropropagated plants	√	−	−	[123]
*Polygonatum odoratum*	Leaves and roots of CC and FC seedlings	√	−	−	[124]
*Bletilla striata*	Leaves, roots, and tubers	−	√	−	[125]
Intra-kingdom stress responses					
*Halostachys caspica*	Salt stress	−	√	−	[126]
*Cicer arietinum*	Ascochyta blight	−	√	−	[127]
*Salvia miltiorrhiza*	Salt stress	−	−	−	[6]
*Astragalus Membranaceus*	Cold stress	−	√	−	[128]
*Zingiber officinale* and *Curcuma amada*	Bacterial wilt	−	√	−	[129]
*Dendrobium huoshanense*	Drought stress	−	√	−	[130]
*Macleaya cordata*	Drought stress	−	√	−	[131]
*Digitalis purpurea*	Cold and dehydration stresses	−	√	−	[75]
*Humulus lupulus*	CBCVd	−	√	−	[15]
*Panax ginseng*	High ambient temperature	−	√	−	[132]
*Aquilaria sinensis*	Wound treatment	−	−	−	[133]
*Panax ginseng*	Dehydration and heat stresses	−	√	−	[134]
*Ziziphus jujuba*	Jujube witches’-broom	−	√	−	[135]
*Polygonatum odoratum*	Consecutive monoculture problem	√	−	−	[124]
*Pogostemon cablin*	Consecutive monoculture problem	−	√	−	[136]
Other research functioning in intra-kingdom					
*Eucommia ulmoides*	First report	−	√	−	[137]
*Taxus*	First report	−	√	−	[138]
*Lotus japonicus*	First report	√	−	−	[139]
*Humulus lupulus*	First report	−	√	−	[140]
*Persicaria minor*	First report	√	−	−	[141]
*Gymnema sylvestre*	First report	√	−	−	[142]
*Rehmannia glutinosa*	First report	√	−	−	[143]
29 medicinal plants	Database	√	−	−	[144]
*Papaver somniferum*	Non-classical miRNA	√	−	−	[18]
*Hypericum*	Evolutionary analysis	√	−	−	[145]
*Pinellia pedatisecta*	Evolutionary analysis	−	−	−	[146]
*Aquilegia coerulea*	Evolutionary analysis	√	−	−	[9]
*Elettaria cardamomum*	Evolutionary analysis	−	√	−	[118]

CBCVd—citrus bark cracking viroid, CC—consecutive cropping, FC—first cropping, GLV—green leaf volatile, GMCs—genetically modified cells, HIV—human immunodeficiency virus, and MAPK—mitogen-activated protein kinase. Indirect verification: studies that miRNA-target gene modules have not been functionally validated at the transgenic level, but the suppression relationship between their miRNAs and target genes has been verified using qRT-PCR, RACE, degradome sequencing, northern blot, β-glucuronidase reporter gene staining (GUS), and/or transient luciferase signal system.

**Table 2 ijms-23-10477-t002:** Names of medicinal plant miRNAs in cross-kingdom and secondary metabolism regulations.

Name of miRNA	Aim Pathway
Cross-kingdom	
**• MIR2911**	**COVID-19, influenza A virus, and tumor proliferation**
**• Gas-miR01, and 02**	**Anti-inflammatory**
**•** MiR414	Alzheimer’s diseases, diabetes, hypoganglionosis, and inflammatory bowel diseases
**•** Oba-miR156f, and 156t	Bile duct carcinoma, lung cancer, and osteoarthritis
**•** Oba-miR160g	Lung cancer, nephronophthisis, and retinitis pigmentosa
**•** Oba-miR482a	Breast cancer, gastric cancer, and ovarian cancer
**•** MiR869.1	Alzheimer’s diseases, cataracts, and diabetes mellitus
**•** Bmn-miR156, 167h, 172d, and 396g	Immune responses
**•** MiR166	Glioblastoma, papillary thyroid carcinoma, and secretory breast carcinomas
**•** Cac-miR-29c-5p	Breast cancer, and ovarian cancer
**•** Cac-miR-4723-3p	Prostate cancer, and renal cancer
**•** Cac-miR-548d-3p, 5653, 5780d, and 7009-3p	Tumor proliferation, ovarian clear cell adenocarcinoma, breast cancer, and lung cancer
**•** MiR10206, 5059, 5073, 5272, 6135, oba-miR531, and aba-miRNA-9497	Tumor proliferation, psoriasis, Alzheimer’s disease, epilepsy syndromes, immune responses, retinitis pigmentosa, and central nervous system toxicity
Secondary metabolism	
**• MiR5298b, and 8154**	**Phenylpropanoid**
**• Smi-miR396b, and miR408**	**Salvianolic acid**
**• MiR5298b, and 8154**	**Taxol**
**• MiR160b, ath-MIR160b and smi-miR396b**	**Tanshinone**
**• MiR156**	**Sesquiterpene**
**•** MiR035, 1168.2, 1438, 156b, 170, 172i, 1858,1873, 2275, 2673a, 2910, 2919, 396b, 408, 5015, 5021, 5658, 828b, 829.1, 8291, f10132-akr, ain-miR1533c, ain-miR156, ain-miR157, cro-miR397a, cro-miR828a, Cs-miR156, mko-miR159b-3p, mko-miR167c-5p, mko-miR168b, mko-miR5082, mko-miR858, mko-miR8610.1, smi-miR12112, smi-miR397, smi-miR396b, and smi-miR408	Phenolic compounds
**•** MiR_116, _1194, _1276, _15, _1508, _1900, _2141, _2596, _334, _853, 1134, 1533, 160, 164, 167a, 167b, 171, 172, 172d-3p, 2919, 396a, 398f/g, -4995, 5563-x, 5021, 5658, 6435, 838, ain-miR1525, dfr-miR156b, dfr-miR160a, mko-miR156, mko-miR167a, mko-miR396c, mko-miR396g-5p, mko-miR5082, mko-miR827b, mko-miR858, novel-m0022-5p, pmi-miR6300, pmi-miR6173, pmi-miR530, and pmi-Nov_13	Terpenoid
**•** MiR159, 159a, 166, 171, 172, 2673a, 390, 396, 858, cro-miR160, EY064998, EY082442, EY107691, EY57163, leaf-miR-477, leaf-miR530, root-miR159, root-miR5140, mko-miR159b-3p, mko-miR5082, mko-miR858, mko-miR8610.1, novel miR_218, novel miR_2432, novel miR2642, novel miR_2924, novel miR_457, and novel miR_853	Esters
**•** MiR2673a, 396, cro-miR160, pso-miR13, pso-miR2161, and pso-miR408	Alkaloids
**•** MiR156, 5298b, 8154, and novel_miR_47	Saponins
**•** MiR5072, MIR1446-x, and MIR394-y	Quinone
**•** MiR156, 414, 5015b, and 5021	Essential oil
**•** NovelmiRNA-191, novelmiRNA-23, and novelmiRNA-58	Triacylglycerols
**•** Pmi-miR396b and pmi-Nov_12	Green leaf volatile
**•** MIR845-y	Steroid
**•** MiR5021	Strictosidine

Bold, miRNAs that have completed functional verification at the level of GMOs or GMCs. Citations refer to Table 1.

## Data Availability

Not applicable.

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
