# Peer review of "MicroRNAs in Medicinal Plants"

_ijms, 2022, doi:10.3390/ijms231810477_

Round 1

Reviewer 1 Report

Dear Authors.

Many plants contain metabolites with biological activities. The development of bioinformatics methods has opened up prospects for determining the mechanisms of formation of the properties and characteristics of metabolites. The scientific content of the manuscript justifies its publication, but some additions and modifications will significantly improve the quality of the article.

Major comments:

1) In Abstract, the potential of using the results should be added.

2) In Introduction, the goal should be formulated.

3) The authors need to describe the methodology for selecting publications and plants for analysis

4) All assumptions about the functionality of the gene or micro-RNA require proof. This should be discussed in conclusion.

5) In the References, 44% of publications refer to 2017-2022 (the last 5 years); the remaining 56% of used sources are older than 5 years. It is recommended to increase the share of references to sources published over the last 5 years when analyzing the current state of research in the area under consideration, since this area of knowledge is rapidly developing.

Author Response

Responses to Reviewer #1

Many plants contain metabolites with biological activities. The development of bioinformatics methods has opened up prospects for determining the mechanisms of formation of the properties and characteristics of metabolites. The scientific content of the manuscript justifies its publication, but some additions and modifications will significantly improve the quality of the article.

[Response]: We appreciate all the comments. We accepted all the suggestions and made corrections accordingly in the revised manuscript.

Comment [1] In Abstract, the potential of using the results should be added.

[Response]: We accepted and added a sentence as “The discussion of the latest results furthers the understanding of medicinal plant miRNAs and helps the rational design of the corresponding miRNA/target genes functional modules.” in Abstract. (Page 1 in the revised manuscript)

Comment [2] In Introduction, the goal should be formulated.

[Response]: In order to formulate a goal of the manuscript, we rewrote a sentence as “Overall, this review summarizes the characteristics of the discipline development of miRNAs in medicinal plants, facilitates the establishment of a complete cross- or intra-kingdom miRNA/target gene module verification system, and discusses the possible challenges of systematic research in the future”. (Page 2 in the revised manuscript)

Comment [3] The authors need to describe the methodology for selecting publications and plants for analysis.

[Response]: The methodology for selecting publications and plants for analysis was added in the last paragraph of Introduction. The sentences added as “The criteria for document search were as follows: (1) database, Web of Science Core Collection, (2) edition, SCI-EXPANDED-1980-present, (3) searched field, All Fields, (4) searched word pairs, medicinal plant + microRNA, herb + microRNA, traditional Chinese medicine + microRNA, and herbaceous plant + microRNA (the “+” means another search line), (5) document types, article, (6) language, English, and (7) searched date, September 5, 2022. Manual search was also used to retrieve relevant papers from the results of automatic search. The papers not included in Journal Citation Reports (JCR) were deleted, and the remaining papers were selected in this study”. (Page 2 in the revised manuscript)

Comment [4] All assumptions about the functionality of the gene or micro-RNA require proof. This should be discussed in conclusion.

[Response]: We appreciate the comment. We discussed the relevant content in Conclusion section, i.e., “Nonetheless, the intra-kingdom functional validation of medicinal plant miRNAs has entered a new era. The technology of genome assembly in non-model plants is becoming mature. In the latest three years, more than 100 articles (over 60%) have been published on genomic information about medicinal plants [91]. Some medicinal plants have published two or more genome versions (e.g., Panax notoginseng [92-96], Andrographis paniculata [97,98], and Gastrodia elata [99,100]), which have significantly supported the miRNA re-search. The research progress of medicinal plant miRNAs will develop rapidly in the next few years. With the enrichment of the cross-kingdom function of miRNAs derived from medicinal plants, their intra-kingdom influence will also be given enough attention. More and more in-depth experiments will be used to provide molecular biology evidence of the predicted candidate miRNA/target gene modules.”. (Page 13 in the revised manuscript)

Comment [5] In the References, 44% of publications refer to 2017-2022 (the last 5 years); the remaining 56% of used sources are older than 5 years. It is recommended to increase the share of references to sources published over the last 5 years when analyzing the current state of research in the area under consideration, since this area of knowledge is rapidly developing.

[Response]: We accepted and added 13 references refer to 2017-2022 (the last 5 years) in the revised manuscript. (Page 17 and 19 in the revised manuscript)

  • Cheng, Q.; Ouyang, Y.; Tang, Z.; Lao, C.; Zhang, Y.; Cheng, C.; Zhou, H. Review on the development and applications of medicinal plant genomes. Plant Sci. 2021, 12, 791219.
  • Chen, W.; Kui, L.; Zhang, G.; Zhu, S.; Zhang, J.; Wang, X.; Yang, M.; Huang, H.; Liu, Y.; Wang, Y. Whole-genome sequencing and analysis of the Chinese herbal plant Panax notoginseng. Plant 2017, 10, 899–902.
  • Zhang, D.; Li, W.; Xia, E.; Zhang, Q.; Liu, Y.; Zhang, Y.; Tong, Y.; Zhao, Y.; Niu, Y.; Xu, J. The medicinal herb Panax notoginsenggenome provides insights into ginsenoside biosynthesis and genome evolution. Plant 2017, 10, 903–907.
  • Fan, G.; Liu, X.; Sun, S.; Shi, C.; Du, X.; Han, K.; Yang, B.; Fu, Y.; Liu, M.; Seim, I. The chromosome level genome and genome-wide association study for the agronomic traits of Panax notoginseng. Iscience 2020, 23, 101538.
  • Jiang, Z.; Tu, L.; Yang, W.; Zhang, Y.; Hu, T.; Ma, B.; Lu, Y.; Cui, X.; Gao, J.; Wu, X. The chromosome-level reference genome assembly for Panax notoginseng and insights into ginsenoside biosynthesis. Plant Commun. 2021, 2, 100113.
  • Yang, Z.; Liu, G.; Zhang, G.; Yan, J.; Dong, Y.; Lu, Y.; Fan, W.; Hao, B.; Lin, Y.; Li, Y. The chromosome–scale high–quality genome assembly of Panax notoginseng provides insight into dencichine biosynthesis. Plant Biotechnol. J. 2021, 19, 869–871.
  • Sun, W.; Leng, L.; Yin, Q.; Xu, M.; Huang, M.; Xu, Z.; Zhang, Y.; Yao, H.; Wang, C.; Xiong, C. The genome of the medicinal plant Andrographis paniculata provides insight into the biosynthesis of the bioactive diterpenoid neoandrographolide. Plant J. 2019, 97, 841–857.
  • Liang, Y.; Chen, S.; Wei, K.; Yang, Z.; Duan, S.; Du, Y.; Qu, P.; Miao, J.; Chen, W.; Dong, Y. Chromosome level genome assembly of Andrographis paniculata. Genet. 2020, 11, 701.
  • Yuan, Y.; Jin, X.; Liu, J.; Zhao, X.; Zhou, J.; Wang, X.; Wang, D.; Lai, C.; Xu, W.; Huang, J. The Gastrodia elata genome provides insights into plant adaptation to heterotrophy. Commun. 2018, 9, 1–11.
  • Chen, S.; Wang, X.; Wang, Y.; Zhang, G.; Song, W.; Dong, X.; Arnold, M.; Wang, W.; Miao, J.; Chen, W. Improved de novo assembly of the achlorophyllous orchid Gastrodia elata. Genet. 2020, 11, 580568.
  • Samad, A.; Rahnamaie-Tajadod, R.; Sajad, M.; Jani, J.; Murad, A.; Noor, N.; Ismail, I. Regulation of terpenoid biosynthesis by miRNA in Persicaria minor induced by Fusarium oxysporum. BMC Genomics 2019, 20, 1–22.
  • Song, C.; Guan, Y.; Zhang, D.; Tang, X.; Chang, Y. Integrated mRNA and miRNA Transcriptome Analysis Suggests a Regulatory Network for UV–B-Controlled Terpenoid Synthesis in Fragrant Woodfern (Dryopteris fragrans). J. Mol. Sci. 2022, 23, 5708.
  • Wu, H.; Noda, N.; Mikami, R.; Kang, X.; Akita, Y. Insertion of a novel transposable element disrupts the function of an anthocyanin biosynthesis-related gene in Echinacea purpurea. Sci. Hortic-Amsterdam. 2021, 282, 110021.

Reviewer 2 Report

This study comprehensively summarized miRNAs in research progress in medicinal plants. The authors categorize medicinal plant miRNA function into levels and collected most of functional research progress of medicinal plant miRNAs. I believe this study would benefit community for studying miRNAs in medicinal plants. But there are some minor points should be addressed.

1. It would be better to give conceptions of cross- or intra-kingdom groups on page 1 line 44-45.

2. On line 57 page 2, what kind of species involved in the comparative genomics approach should be mentioned.

3. For legend of Figure 2. I do not agree this is specific for the medicinal plants. If it is, what’s the specific components in the pathway compared to the other eukaryotic species.

4. On line 88 page 3, I am a little confused about the authors using ‘MIRNAs’ in this place but using ‘miRNAs’ in other places. It would be better to allow them to be consistent.

5. On line 114 page 4, It would be better to add a simple graph to show the non-canonical pathway.

6. On line 257 page 10. It can be improved to discuss a little about the reproductive stage like flowering. I know Galla et al. 2013 did some works relating to flowering transcriptome of Hypericum perforatum.

Author Response

Responses to Reviewer #2

This study comprehensively summarized miRNAs in research progress in medicinal plants. The authors categorize medicinal plant miRNA function into levels and collected most of functional research progress of medicinal plant miRNAs. I believe this study would benefit community for studying miRNAs in medicinal plants. But there are some minor points should be addressed.

[Response]: We gratefully appreciate for your valuable comments above and below.

Comment [1] It would be better to give conceptions of cross- or intra-kingdom groups on page 1 line 44-45.

[Response]: We agree with the comment and believe that the concept of cross- or intra-kingdom groups is very important to this review article. Actually, the conceptions of cross- or intra-kingdom groups has been given in the Abstract of the original manuscript. If necessary, we will add this concept into the introduction.

Comment [2] On line 57 page 2, what kind of species involved in the comparative genomics approach should be mentioned.

[Response]: The sentence only referd to the corresponding genus “aquilegia” but did not specify the Latin name of the species. The Latin name (Aquilegia coerulea) was added in the sentence as “Comparative genomics approach identified 45 Aquilegia coerulea miRNAs that target genes involved in metabolism and stress responses”. (Page 2 in the revised manuscript)

Comment [3] For legend of Figure 2. I do not agree this is specific for the medicinal plants. If it is, what’s the specific components in the pathway compared to the other eukaryotic species.

[Response]: We apologized for our negligence of the mistake. The legend of Figure 2 has been modified as “Canonical miRNA-generating pathway in plant kingdom”. (Page 3 in the revised manuscript)

Comment [4] On line 88 page 3, I am a little confused about the authors using ‘MIRNAs’ in this place but using ‘miRNAs’ in other places. It would be better to allow them to be consistent.

[Response]: “MIRNA” refers to “precursor miRNA”, while “miRNA” refers to “mature miRNA”. MiRNA is modified from MIRNA. Therefore, they are different RNA molecules.

Comment [5] On line 114 page 4, It would be better to add a simple graph to show the non-canonical pathway.

[Response]: Study in the animal kingdom have explained the possible non-canonical synthesis pathway of miRNA (miRNA ID: miR–712, article title: The atypical mechanosensitive microRNA–712 derived from pre–ribosomal RNA induces endothelial inflammation and atherosclerosis). However, miRNAs from the non-canonical pathway have been found to be very limited in the plant kingdom, and their synthetic pathways are still unclear. Considering that there is not enough and reliable literature to support the visualization of non-canonical pathway, we still retain the existing canonical pathway to highlight the theme of the manuscript.

Comment [6] On line 257 page 10. It can be improved to discuss a little about the reproductive stage like flowering. I know Galla et al. 2013 did some works relating to flowering transcriptome of Hypericum perforatum.

[Response]: Thank you for your helpful comment. The article of Galla et al. 2013 you mentioned was listed in Table 1 of the original manuscript, but it was not discussed. In the revised manuscript, we added a sentence as “The Hypericum perforatum flowers shared highly conserved miRNAs and that these miRNAs potentially target functional genes involved in stress response, flower development and plant reproduction”. (Page 11 in the revised manuscript)

Round 2

Reviewer 1 Report

Dear Authors

My comments are taken into account